# Facilitators and barriers of relatives' involvement in nursing care decisions and self-care of patients with acquired brain injury or malignant brain tumour: A scoping review protocol

**Rikke Guldager**[1☯]*, **Mia Ingerslev Loft**[2☯], **Sara Nordentoft**[1☯], **Lena Aadal**[3☯], **Ingrid Poulsen**[4☯]

**1** Neurosurgical Department, Rigshospitalet, Copenhagen, Denmark, **2** Department of Neurology, Rigshospitalet, Copenhagen, Denmark, **3** Hospitalsenhed Midt, Hammel Neurorehabilitation and Research Centre, Hammel, Denmark, **4** Department of Neurorehabilitation, TBI Unit, Rigshospitalet, Hvidovre, Denmark

☯ These authors contributed equally to this work.
* rikke.guldager.01@regionh.dk

**Data Availability Statement:** No datasets were generated or analysed during the current study. All

## Abstract

### Introduction

Involving relatives can contribute to better quality of care and treatment, and lead to higher satisfaction with hospitalisation in the patients, relatives and healthcare professionals. Nurses play an important role in developing a trusting relationship and facilitating relatives' involvement, since they are around the patient and relatives all day. Thus, involvement is central to the nursing practice. However, involving relatives seems complex and multifaceted with many possible facilitators and barriers to nurses.'

### Objective

The objective of this scoping review is to identify and map the available evidence concerning possible facilitators and barriers to nurses involving relatives in the course of treating disease in individuals who have sustained an acquired brain injury or malignant brain tumour in all settings.

### Methods and analysis

The proposed scoping review will be performed following the Joanna Briggs Institute's (JBI) methodology for scoping reviews. Indexed and grey literature in English, Scandinavian or German languages from 2010 to the present will be considered. The searches will be conducted using bibliographic databases: Medline (via PubMed), CINAHL (via EBSCO) and EMBASE (via OVID). Two reviewers will independently screen the studies and determine if their title, abstract and full text meet the study's inclusion criteria. In case of disagreement, a third and fourth reviewer will be consulted. A customised data extraction form will be used to extract data from the included studies. The results will be presented in tabular form,

relevant data from this study will be made available upon study completion.

**Funding:** Main author received the funding This study is supported by The Danish Health Confederation and Danish Regions (Grant number 2657). The funding body will not have any role in the review process.

**Competing interests:** The authors have declared that no competing interests exist.

accompanied by a narrative summary related to the objective of the present scoping review. This scoping review will consider studies that involve nurses caring for individuals with an acquired brain injury or malignant brain tumour in all settings (community, primary care, health care centres, hospital and long-term care institutions). Studies will be included if they describe any kind of facilitators or barriers to involving relatives, and the review will consider all study designs.

## Introduction

It is estimated that the incidence for traumatic brain injury (TBI) in the US and Europe is 30 per 100,000 persons per year, and globally, stroke affects an estimated 17 million people per year [1]. Acquired brain injury (ABI) is an injury to the brain that has occurred after birth but is not related to congenital defect or degenerative disease. The origin of the injury can be traumatic or non-traumatic (e.g. stroke) [2]. The peak age of TBI is between the ages of 15–24 years and 75 years and older [3]. Patients with severe ABI are often affected in several areas (functional, health, personal and environmental) that impair their ability to collaborate with health care professionals (HCPs) and participate in making decisions about their own treatment, rehabilitation and/or future. Rehabilitation of patients with ABI, therefore, differs from rehabilitation of many other patient groups and the collaboration with their relatives is particularly important [4]. Malignant brain tumour (MBT) often occurs between the ages of 40–70 years, and has an estimated incidence in adults of 7.3 cases per 100,000 persons [5, 6]. Patients with MBT develop symptoms for months to days due to the mass effect of the tumour itself and/or additional brain swelling caused by oedema. The most common general symptoms are headache, nausea and fatigue. Later, the general symptoms can worsen into vomiting, balance difficulties, epileptic seizures and various cognitive problems [7]. The patient's path is often characterised by several examinations within a short time span, from referral by a general practitioner to diagnostic work-up and initial treatment [8]. As a consequence, they experience many contacts with different hospital departments and HCPs, often in different hospitals, in a time filled with waiting, anxiety, loss of control and uncertainty for the future [9].

Research has shown that patients with MBT need psychosocial support interventions that address their individual needs, a network to relieve them and for the HCPs to recognise these support systems as an important resource [10]. Thus, sustaining an ABI or a MBT has major consequences for patients and their relatives, and has shown to result in familial strain.

Involving patients and their relatives in their treatment and care is required according to existing health policies in most western countries [11]. Unfortunately, there is a range of different interpretations of how and to which patients and relatives can be involved in their treatment and the decision-making process. Further, research literature refers to different definitions and terminologies as partnership, involvement and shared decision-making [12]. In the instant protocol, the term involvement will be used, as it refers to an active doing on the individual level [12]. It is well-known that effective involvement requires that HCPs have sufficient knowledge of patients' and relatives' wishes for involvement, and that patients and relatives are genuinely able to influence decision-making processes [13]. Furthermore, trust and respect between HCPs and patients/relatives are important for participating in making decisions about care [13].

Nurses are constantly around patients and relatives and, therefore, are well-positioned to establish a trustful relationship and ensure safe, affordable and respectful care [14].

Establishing a trusting relationship includes five core elements, explicitly through communication by the nurse to the patient/relative: focusing, knowing, trusting, anticipating and evaluating [15]. Focusing nursing care on establishing a trusting relationship with patients and relatives in the initial stage of hospitalisation may contribute to improved practices of involvement in the rehabilitation process, where such involvement is based on identifying the relative's needs and where support could thereby be delivered in a more tailored manner [16].

Another crucial aspect to successful involvement is understanding that relatives and nurses have different spheres of expertise. The nurse is an expert in nursing, while the relatives, in most cases, must be considered experts in the patient's life, able to contribute important, person-specific knowledge of the patient that the nurses do not possess. Previous research has illuminated different facilitators and barriers for involvement. Guldager et al. found relatives' differential and unequal resources function as facilitators and barriers [16]. Facilitators for involvement include participating in nursing care situations, the possibility of being present during hospitalisation, the relationship with the providers, experience with illness, dedication and proactivity [16]. On the contrary, being reactive, not participating in nursing care situations, being unable to express one's own wants and needs and minimal flexibility from the patient's workplace are barriers to involvement [16]. Fisher et al. suggest increasing relatives' competences in order to address unmet informational and practical support needs that they might have, but also to benefit individuals with a brain injury by optimising clinical outcomes. However, relatives of patients with a TBI or MBT often suffer from feelings of anxiety and depression, which may potentially create barriers to involvement because of decreased energy and capacity to positively impact the process [17]. Keatinge et al. found that patients considered communication to be the principal barrier to successful partnerships between patients and relatives and concluded that nurses' lack of communication skills was a barrier to involvement [18]. Lastly, the organisational and work environment can negatively influence patient and relative involvement with lack of time being identified as a barrier [13].

Despite the fact that research shows that involving relatives contributes to better treatment for patients [19], affects their psychological and emotional well-being [20], and ultimately has a positive impact on the safety of patient care [20], it still seems like there are many facilitators and barriers to involvement. Further, it is unclear if the HCPs have the competencies to meet the relatives' diverse needs and preferences for involvement, or if involvement is just a political ideal.

A preliminary search for existing scoping and systematic reviews on the topic has been conducted in the Cumulative Index to Nursing and Allied Health Literature (CINAHL) and in PubMed in February 2021. No relevant completed or ongoing systematic or scoping reviews were found.

## Aim and research question

The objective of this scoping review is to identify and map the available evidence on possible facilitators and barriers to nurses' involvement of relatives throughout the course of the patients' ABI or MBT, in all settings.

This scoping review's research question is: what kind of facilitators and barriers do nurses face in involving the relatives of individuals with ABI or MBT?

## Methods

We will conduct a scoping review to identify and map the evidence concerning involving relatives of patients with ABI or MBT in accordance with the Joanna Briggs Institute's (JBI) methodology for scoping reviews [21]. A scoping review is suitable for our topic area as it is defined

as 'a form of knowledge synthesis that addresses an exploratory research question aimed at mapping key concepts, types of evidence and gaps in research related to a defined area or field by systematically searching, selecting and synthesizing knowledge [21]. In our scoping review, at least two independent reviewers will participate in an iterative process of screening the literature, paper selection and data extraction. Disagreements between the reviewers will be resolved by discussion until consensus is reached or after consultation with the research team, when needed. Results will be reported with descriptive statistics and diagrammatic or tabular displayed information, accompanied by narrative summaries as explained in the JBI guidelines [22].

## Inclusion criteria

**Population.** This review will include studies that involve registered nurses caring for individuals with an ABI or MBT and their relatives in all settings.

**Concept.** The overarching concept of interest for the scoping review is the kind of possible facilitators and barriers that nurses face when involving the relatives of patients with ABI or MBT in all settings. We expect to identify text representing both concepts of facilitators and barriers independently; but also, a relationship between the two. Thus, this review will consider studies that provide information about how nurses experience involving relatives of patients with ABI or MBT in their nursing care. Articles focusing on nurses caring for patients with neurological diseases such as Parkinson's, Alzheimer's, dementia or multiple sclerosis will be excluded.

**Context.** This review will consider studies that involve possible facilitators and barriers towards nurses' involvement of relatives of individuals with ABI or MBT in all settings. The review has no limitations to a particular country or healthcare system.

**Eligible study designs and studies.** This scoping review will consider all types of quantitative, qualitative or mixed-methods studies, reports or theses describing facilitators and barriers to nurses involving the relatives of individuals with ABI or MBT. Grey literature that includes this information, including expert opinions and editorials, will also be included.

**Databases and additional sources.** We will search Medline (via PubMed), CINAHL (via EBSCO) and EMBASE (via OVID).

**Search strategy.** A three-step search strategy will be used. An initial search of the databases PubMed, CINAHL and EMBASE for facilitators and barriers to nurses' involving relatives of individuals with ABI or MBT will be conducted. Afterwards, an analysis was conducted of text in both the title and abstract of retrieved articles and of the index terms used to describe the articles. Key terms were determined through discussions between two authors (RG and IP) and a university hospital librarian.

A second search using all keywords and index terms will be undertaken across all relevant databases. The search will be iterative as reviewers become more familiar with the evidence bases. Additional useful keywords, sources and search terms will be incorporated into the search strategy. Consultation with a university hospital librarian will guide the search's design and refinement. The search will use keywords and Medical Subject Headings (MeSH) terms (Table 1). The search strategies will be created specifically for each database using relevant index and free text terms.

The search will cover studies published in English, Scandinavian or German from January 2010 to the present in order to ensure that included reports are relevant to current clinical practice and legislation. A full search strategy for PubMed database is provided (Table 2).

In the third step, the reference list of identified reports and articles will be appraised and screened for additional studies. The titles and abstracts of all identified studies that are

**Table 1. Search terms.**

| Participants/population | Concept | Context |
|---|---|---|
| **Nurse** | Barriers | **Brain injuries** |
| Registered nurse | Facilitators | Acquired brain injury |
| Practice nurse | Motivation | Stroke |
| License nurses | Participation | **Brain Neoplasms** |
| | Patient participation | Glioblastoma |
| | Involvement | |
| | Family needs | |
| | Decision making, shared | |
| | Shared decision making | |
| | Care | |
| | Caring | |
| | Neuroscience nursing | |
| | Nurses role | |
| | Patient participation | |
| | Family practice | |

potentially eligible for inclusion in the review will be screened, and full-text versions of included articles will be obtained.

All databases will be exported into EndNote X8.1 software (Clarivate Analytics, PA, USA). Duplicates will be removed before each entry is screened for eligibility. Then, all of the titles and abstracts of the retrieved studies will be uploaded to the Covidence systematic review software (Veritas Health Innovation Ltd, Melbourne, Australia) for screening [23].

**Study selection.** Study selection will be conducted in two stages. In the first stage, five independent reviewers (RG, SN, LA, IP and MI) will screen the titles and abstracts against the inclusion criteria. In the second stage, the entire research team will conduct a final review of all potentially relevant, full-text articles that are retrieved and screened for inclusion. Any disagreements will be resolved through discussion and consensus with the entire research team. The systematic literature search will be summarised and presented in a PRISMA-ScR flow chart as suggested by JBI [22].

**Data extraction.** As recommended by JBI [22] the data extraction will involve two independent reviewers using a draft data extraction. Expected extraction fields will include:

**Table 2. Initial search strategy for Medline (via PubMed).**

| Search | Query | Records retrieved |
|---|---|---|
| #1 Participants/ Population | **((((Nurse) OR (nurse)) OR (registered nurse)) OR (practice nurse)) OR (license nurse)** | 163,202 |
| #2 Content | ((((((((((((((Barriers)) OR (Facilitators)) OR (Motivation)) OR (Participation)) OR (Patient participation)) OR (Involvement)) OR (Family needs)) OR (Decision making)) OR (Shared decision making)) OR (care)) OR (caring)) OR (Neuroscience nursing)) OR (Patient participation)) OR (Family practice) | 397,920 |
| #3 Context | ((((brain injury) OR (Acquired brain injury)) OR (Stroke)) OR (Brain Neoplasms)) OR (Glioblastoma) | 39,728 |
| #4 | ((((((brain injury) OR (Acquired brain injury)) OR (Stroke)) OR (Brain Neoplasms)) OR (Glioblastoma) AND (2010/1/ 1:2021/9/30[pdat])) AND ((((((((((((((() OR (Barriers)) OR (Facilitators)) OR (Motivation)) OR (Participation)) OR (Patient participation)) OR (Involvement)) OR (Family needs)) OR (Decision making)) OR (Shared decision making)) OR (care)) OR (caring)) OR (Neuroscience nursing)) OR (Patient participation)) OR (Family practice) AND (2010/1/1:2021/9/30 [pdat]))) AND (((((Nurse) OR (nurse)) OR (registered nurse)) OR (practice nurse)) OR (license nurse) AND (2010/1/ 1:2021/9/30[pdat])) | 212 |

**Table 3. Data presentation template.**

| Research question: What kind of possible facilitators and barriers towards nurses' involvement of relatives through the course of disease of individuals with ABI or MBT in all settings? | | | | | | | |
|---|---|---|---|---|---|---|---|
| Author/year of publication/ country of origin | Aim | Methods | Design | Characteristics of facilitators and barriers for involvement | | Context | Key findings |
| | | | | Facilitators | Barriers | | |

- Author(s)

- Year of publication

- Origin/country of origin

- Setting

- Study population

- Methodology/methods

- Facilitators to involvement of relatives from the nurses' perspectives

- Barriers to involvement of relatives from the nurses' perspectives

- Context

The data extraction tool will be pilot tested on three articles, and the team will discuss and decide on any required revisions of the tool (after testing). Any modifications will be detailed in the final scoping review. A third and fourth reviewer will resolve any disagreements between the reviewers. Authors of studies will be contacted to request missing or additional data, where required.

**Presentation of the results.** As suggested by JBI, the extracted data will be presented in a diagrammatic or tabular form in a manner that aligns with the objective of this scoping review [22] to identify and map the available evidence concerning possible facilitators and barriers to nurses' involvement of relatives of individuals with ABI or MBT in all settings. A narrative summary will accompany the tabulated and/or charted results, which will describe how the results relate to the review's question (Table 3) [22]. A data presentation template will be developed specifically for this scoping review.

## Acknowledgments

The authors would like to acknowledge the university hospital librarian, Karine Korsgaard, for her contribution with conceptualising key terms.

## Author Contributions

**Conceptualization:** Rikke Guldager, Mia Ingerslev Loft, Sara Nordentoft, Lena Aadal, Ingrid Poulsen.

**Formal analysis:** Rikke Guldager, Mia Ingerslev Loft, Lena Aadal, Ingrid Poulsen.

**Funding acquisition:** Rikke Guldager.

**Methodology:** Rikke Guldager, Mia Ingerslev Loft, Sara Nordentoft, Lena Aadal, Ingrid Poulsen.

**Project administration:** Rikke Guldager.

**Resources:** Rikke Guldager.

**Validation:** Rikke Guldager, Mia Ingerslev Loft, Sara Nordentoft, Lena Aadal, Ingrid Poulsen.

**Visualization:** Rikke Guldager.

**Writing – original draft:** Rikke Guldager, Mia Ingerslev Loft, Sara Nordentoft, Lena Aadal, Ingrid Poulsen.

**Writing – review & editing:** Rikke Guldager, Mia Ingerslev Loft, Sara Nordentoft, Lena Aadal, Ingrid Poulsen.

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
