## [Decision Letter · Decision Letter 0]

25 Jan 2022

PONE-D-21-20624

Facilitators and barriers towards nurses’ involvement of relatives: a scoping review protocol

PLOS ONE

Dear Dr. Guldager,

Thank you for submitting your manuscript to PLOS ONE. After careful consideration, we feel that it has merit but does not fully meet PLOS ONE’s publication criteria as it currently stands. Therefore, we invite you to submit a revised version of the manuscript that addresses the points raised during the review process. 

The manuscript has been evaluated by three reviewers, and their comments are available below.

The reviewers have raised a number of concerns that need attention. In particular, reviewer 1 is a stats reviewer and he has several comments about the reporting of the study, methodological aspects of the study , revisions to the statistical analyses that need to be addressed. Could you please revise the manuscript to carefully address the concerns raised by all reviewers?

We look forward to receiving your revised manuscript.

Kind regards,

Elisa Panada

Associate Editor

PLOS ONE

Journal Requirements:

5. Please amend your authorship list in your manuscript file to include authors Rikke Guldager, Mia Ingerslev Loft, Sara Nordentoft, Lena Aadal, and Ingrid Poulsen.

6. Please amend your list of authors on the manuscript to ensure that each author is linked to an affiliation. Authors’ affiliations should reflect the institution where the work was done (if authors moved subsequently, you can also list the new affiliation stating “current affiliation:….” as necessary).

8.  We noticed you have some minor occurrence of overlapping text with the following previous publications, which needs to be addressed:

- https://journals.lww.com/jbisrir/Fulltext/2020/04000/Identifying_and_managing_frailty_in_Brazil__a.11.aspx

- https://onlinelibrary.wiley.com/doi/full/10.1002/nop2.326

In your revision ensure you cite all your sources (including your own works), and quote or rephrase any duplicated text outside the methods section. Further consideration is dependent on these concerns being addressed.

Reviewers' comments:

Reviewer's Responses to Questions

**Comments to the Author**

1. Does the manuscript provide a valid rationale for the proposed study, with clearly identified and justified research questions?

Reviewer #1: Yes

Reviewer #2: Yes

Reviewer #3: Yes

2. Is the protocol technically sound and planned in a manner that will lead to a meaningful outcome and allow testing the stated hypotheses?

Reviewer #1: No

Reviewer #2: Partly

Reviewer #3: Yes

3. Is the methodology feasible and described in sufficient detail to allow the work to be replicable?

Reviewer #1: Yes

Reviewer #2: Yes

Reviewer #3: Yes

4. Have the authors described where all data underlying the findings will be made available when the study is complete?

Reviewer #1: Yes

Reviewer #2: No

Reviewer #3: Yes

5. Is the manuscript presented in an intelligible fashion and written in standard English?

Reviewer #1: Yes

Reviewer #2: No

Reviewer #3: Yes

6. Review Comments to the Author

You may also provide optional suggestions and comments to authors that they might find helpful in planning their study.

Reviewer #1: Dear Editor,

Thank you for the opportunity to provide a review of Manuscript PONE-D-21-20624 entitled "Facilitators and barriers towards nurses’ involvement of relatives: a scoping review protocol". My comments relate primarily to the adequacy of the implementation and reporting of epidemiologic and statistical procedures.

The quality of the technical English is appropriate and did not affect my understanding of the manuscript.

# Major Issues

*IMPORTANT* Scoping reviews are underdone systematic reviews. This is seen as an excuse to cut corners and provide substandard processes, all in an effort to privilege speed, cost, or some other non-pertinent reason. A review is either performed systematically or it is not. In the present case, the protocol misses out on several important and crucial methodological steps. If followed, the resulting research will be substandard and inutile.

First, the authors provide no process of synthesis. What theories or frameworks guide their assessment of the evidence? Without a process of synthesis, then the resulting information is no better than a tally or a list of facilitators and barriers. This is hardly research, since the authors will simply be regurgitating information found in the feeder studies. A prime example of this is the dummy table provided by the authors as a guide to the presentation of results. This table is trivial as it simply lists each item tallied by the authors. There is no effort to apply a framework to the results. The nursing literature is replete with significant frameworks, theories or models on which to synthesise the results. That the authors have not cited any is quite disturbing.

Second, the authors do not provide a method of assessing the quality of the studies. The assumption they are using, in effect, is that all the studies were conducted equally well. This is ridiculous.

Third, the authors do not provide a method of understanding the degree of differences between the studies. An example of such differences is context. The involvement of relatives by nurses is highly contextual. For example, Hospital A might have existing programs designed to involve relatives in patient care. Thus, data from Hospital A will list a set of facilitators and barriers that will be quite different from other settings and will be quite inapplicable to others because it depends on the prior existence of such programs. The existence of such environmental factors have to do with the context of the research. This is undefined and ignored.

I am unable to support the approval of this manuscript for publication in the journal.

Thank you.

Reviewer #2: The title is incomplete.. involvement of relatives in what?

The abstract is not organized. you start with introduction, then objectives. inclusion criteria should be part of the methods.

Write in full first before abbreviating - see under introduction, line 1, TBI, ABI,

Consider using the term participation rather than involvement. relatives may be involved but not participate in the patient care.

Conference abstracts and papers and reports regarding policies and strategies in use by professional bodies or organizations will be excluded, please give reason for excluding them.

Methods - indicate the search terms and the criteria you will use to appraise the selected studies.

Table 1: The mesh search terms are not exhaustive. consider adding registered nurses, registered nurse, practice nurse, license nurses. Also for involvement consider adding participation, communication, decision making, care, caring

Reviewer #3: The scoping review protocol was generally good and presented in line with the PRISMA checklist for scoping review(Tricco et al., 2018). However, the reviewer may consider making changes to the title to include or account for the disease condition highlighted in the body of the scoping review as the participants aspect of the PICO framework. This would enhance title completion and improve readers’ experience. There is still, not a mention of the use of critical appraisal tool for quality appraisal. Also, there is no evidence of the data search flow chart in the protocol. This can improve the pictorial or construct validity of the scoping review. The mention of search databases, the dates of publication of interest, and the key search term provided offers readers insight into how search outcome result where obtained: this is good practice needed as prove of reliability. It also means that research outcome can be reproduced elsewhere. Otherwise, the registered protocol is well written in a clear, simple, and concise manner for readers' understanding.

Reference scoping review checklist

Tricco, AC, Lillie, E, Zarin, W, O'Brien, KK, Colquhoun, H, Levac, D, Moher, D, Peters, MD, Horsley, T, Weeks, L, Hempel, S et al. PRISMA extension for scoping reviews (PRISMA-ScR): checklist and explanation. Ann Intern Med. 2018,169(7):467-473. doi:10.7326/M18-0850.

7. PLOS authors have the option to publish the peer review history of their article (what does this mean?). If published, this will include your full peer review and any attached files.

Reviewer #1: No

Reviewer #2: **Yes: **Dr. Haddy Tunkara Bah

Reviewer #3: **Yes: **Patience James

---

## [Author Response · Author response to Decision Letter 0]

17 Feb 2022

Title of the Manuscript: Facilitators and barriers towards nurses’ involvement of relatives: a scoping review protocol

Manuscript Number: PONE-D-21-20624

Reviewer 1

Comment 1: *IMPORTANT* Scoping reviews are underdone systematic reviews. This is seen as an excuse to cut corners and provide substandard processes, all in an effort to privilege speed, cost, or some other non-pertinent reason. A review is either performed systematically or it is not. In the present case, the protocol misses out on several important and crucial methodological steps. If followed, the resulting research will be substandard and inutile.

Authors’ Response: Thank you for the comment. However, we argue as Munn and Levac that scoping reviews are useful for exploring emerging evidence when it is still unclear what other, more specific questions can be posed and valuably addressed by a more precise systematic review (1, 2). We agree that a scoping review need to be rigorously conducted, transparent and trustworthy. To ensure this we refer to the JBI’s Guidance for the conduct of a Scoping Review(3). We have rewritten the method section to make this clearer. 

Change to Text: 

We will conduct a scoping review to identify and map the evidence concerning the involvement of relatives in patients with ABI or MBT in accordance with the Joanna Briggs Institute’s (JBI) methodology for scoping reviews. A scoping review is suitable for our topic area as scoping reviews is defined as ‘a form of knowledge synthesis that addresses an exploratory research question aimed at mapping key concepts, types of evidence, and gaps in research related to a defined area or field by systematically searching, selecting and synthesizing knowledge’ .(4) In our scoping review at least two independent reviewers will participate in an iterative process of screening the literature, paper selection and data extraction. Disagreements between the reviewers will be resolved by discussion until consensus is reached or after consultation with the research team if needed. Results will be reported with descriptive statistics and diagrammatic or tabular displayed information, accompanied by narrative summaries as explained in the JBI guidelines (ref).

Comment 2: First, the authors provide no process of synthesis. What theories or frameworks guide their assessment of the evidence? Without a process of synthesis, then the resulting information is no better than a tally or a list of facilitators and barriers. This is hardly research, since the authors will simply be regurgitating information found in the feeder studies. A prime example of this is the dummy table provided by the authors as a guide to the presentation of results. This table is trivial as it simply lists each item tallied by the authors. There is no effort to apply a framework to the results. The nursing literature is replete with significant frameworks, theories or models on which to synthesise the results. That the authors have not cited any is quite disturbing.

Authors’ Response: 

Thank you for your comment. Please see the comment above. 

Comment 3: Second, the authors do not provide a method of assessing the quality of the studies. The assumption they are using, in effect, is that all the studies were conducted equally well. This is ridiculous.

Authors’ Response: 

Thank you for your comment. According to Munn et.al (2018) It is not within the remit of a scoping review to assess quality of the study or to produce a critically appraised answer to a particular question(1). As Munn states ‘An assessment of methodological limitations or risk of bias of the evidence included within a scoping review is generally not performed’(1). The scoping review methodology is chosen because it is suitable when the aim is to identify and map the available evidence concerning on what kind of possible facilitators and barriers towards nurses’ involvement of relatives to individuals with acquired injury or malignant brain tumour.

Comment 4: 

Third, the authors do not provide a method of understanding the degree of differences between the studies. An example of such differences is context. The involvement of relatives by nurses is highly contextual. For example, Hospital A might have existing programs designed to involve relatives in patient care. Thus, data from Hospital A will list a set of facilitators and barriers that will be quite different from other settings and will be quite inapplicable to others because it depends on the prior existence of such programs. The existence of such environmental factors have to do with the context of the research. This is undefined and ignored.

Authors’ Response: 

Thank you for your comment. As we are interested in what kind of possible facilitators and barriers towards nurses’ involvement of relatives through the course of disease to individuals with acquired injury or malignant brain tumour we will include studies from all settings. To be able to identify possible environmental factors that may influence facilitators and barriers for involvement we have added context to expected extraction fields on P 5. L 180 and to the data presentation template and will extract data related to context/setting (ICU, in-outpatient clonic, primary health care, nursing home ect.). 

Reviewer 2

Comment 1: The title is incomplete.. involvement of relatives in what?

Authors’ Response: Thank you for this comment. We have been more specific about the ‘what’ 

Change to Text: Facilitators and barriers towards nurses’ involvement of relatives in decision making and daily life activities through the course of disease of patients with an acquired brain injury or malignant brain tumour.: a scoping review protocol

Comment 2: The abstract is not organized. you start with introduction, then objectives. inclusion criteria should be part of the methods.

Authors’ Response: Thank you for this comment. The abstract is now organized

Change to Text: 

Comment 3: Write in full first before abbreviating - see under introduction, line 1, TBI, ABI,

Author’s Response: Thank you for pointing this out. This have been corrected on line 1 and line 35

Change to Text: 

Comment 4: Consider using the term participation rather than involvement. relatives may be involved but not participate in the patient care.

Author’s Response: Thank you for your suggestion. As stated on P. 2, line 54 we use the terminology involvement, as it refers to an active doing on the individual level. As we are not only interest in involvement understood as participation in e.g. patient care, but involvement through the course of disease, we maintain the term involvement. 

Change to Text: 

Comment 5: 

Conference abstracts and papers and reports regarding policies and strategies in use by professional bodies or organizations will be excluded, please give reason for excluding them.

Author’s Response: Thank you for your comment. In the methods sections Eligible study designs and studies (P: 4)

we have written that grey literature will be included. Since conference abstracts and papers regarding policies and strategies can be considered as grey literature, we have deleted the sentence

Change to Text: 

Comment 6: Methods - indicate the search terms and the criteria you will use to appraise the selected studies.

Author’s Response: Thank you for your comment. We have created a table 1, that shows the search terms and where the MesH terms are highlighted in bold.

Change to Text: 

Comment 7: Table 1: The mesh search terms are not exhaustive. consider adding registered nurses, registered nurse, practice nurse, license nurses. Also for involvement consider adding participation, communicateon, decision making, care, caring

Author’s Response: Thank you for your suggestions. We have added the suggested terms and have updated the initial search (See table 2)

Change to Text: N/A

Reviewer 3

Comment 1: The scoping review protocol was generally good and presented in line with the PRISMA checklist for scoping review (Tricco et al., 2018). However, the reviewer may consider making changes to the title to include or account for the disease condition highlighted in the body of the scoping review as the participants aspect of the PICO framework. This would enhance title completion and improve readers’ experience.

Author’s Response: Thank you for your comment. We have included the disease conditions as suggested. 

Change to Text: Facilitators and barriers towards nurses’ involvement of relatives through the course of disease of individuals with an acquired brain injury or malignant brain tumour.: a scoping review protocol

Comment 2: There is still, not a mention of the use of critical appraisal tool for quality appraisal. Also, there is no evidence of the data search flow chart in the protocol. This can improve the pictorial or construct validity of the scoping review.

Author’s Response: Thank you for your comment. According to Munn et.al (2018) It is not within the remit of a scoping review to assess quality of the study or to produce a critically appraised answer to a particular question(1). As Munn states ‘An assessment of methodological limitations or risk of bias of the evidence included within a scoping review is generally not performed’(1). The scoping review methodology is chosen because it is suitable to give an indication of the amount of literature as well as and studies available and further to clarify concepts(1). On page 5 we write that the systematic literature search will be summarized and presented in a PRISMA-ScR flow chart as suggested by JBI. (3)

Comment 3: The mention of search databases, the dates of publication of interest, and the key search term provided offers readers insight into how search outcome result where obtained: this is good practice needed as prove of reliability. It also means that research outcome can be reproduced elsewhere

Author’s Response: Thank you for your comment. We have inserted a Table 1, that shows search key search terms. In addition, we have provided information on databases and additional sources on page 4, line 136-137 and dates of publication of interest on page 5, line 152-154

Comment 4: 

Otherwise, the registered protocol is well written in a clear, simple, and concise manner for readers' understanding.

Author’s Response: Thank you very much for this comment

1. Munn Z, Peters MDJ, Stern C, Tufanaru C, McArthur A, Aromataris E. Systematic review or scoping review? Guidance for authors when choosing between a systematic or scoping review approach. BMC Medical Research Methodology. 2018;18(1):143.

2. Levac D, Colquhoun H, O'Brien KK. Scoping studies: advancing the methodology. Implementation Science. 2010;5(1):69.

3. Peters MDJ GC, McInerney P, Munn Z, Tricco AC, Khalil, H. . Chapter 11: Scoping Reviews. . In: Aromataris E MZ, editor. JBI Manual for Evidence Synthesis. Adelaide: JBI2020.

4. Colquhoun HL, Levac D, O'Brien KK, Straus S, Tricco AC, Perrier L, et al. Scoping reviews: time for clarity in definition, methods, and reporting. J Clin Epidemiol. 2014;67(12):1291-4.

---

## [Decision Letter · Decision Letter 1]

5 Jul 2022

PONE-D-21-20624R1Facilitators and barriers towards nurses’ involvement of relatives in decision making and daily life activities through the course of disease of individuals with an acquired brain injury or malignant brain tumour: a scoping review protocolPLOS ONE

Dear Dr. Guldager,

I have received the reviews from 2 experts and they agree that your paper has improved. However, as you will see below Reviewer #2 points to the issue of appraisal of studies. Before I will consider your paper for publication, please take notice of this remark and try to address it either in a rebuttal or in a revised paper.

We look forward to receiving your revised manuscript.

Kind regards,

Robert Didden

Academic Editor

PLOS ONE

Journal Requirements:

Reviewers' comments:

Reviewer's Responses to Questions

**Comments to the Author**

1. Does the manuscript provide a valid rationale for the proposed study, with clearly identified and justified research questions?

Reviewer #2: Yes

Reviewer #3: Yes

2. Is the protocol technically sound and planned in a manner that will lead to a meaningful outcome and allow testing the stated hypotheses?

Reviewer #2: Partly

Reviewer #3: Yes

3. Is the methodology feasible and described in sufficient detail to allow the work to be replicable?

Reviewer #2: Yes

Reviewer #3: Yes

4. Have the authors described where all data underlying the findings will be made available when the study is complete?

Reviewer #2: Yes

Reviewer #3: Yes

5. Is the manuscript presented in an intelligible fashion and written in standard English?

Reviewer #2: Yes

Reviewer #3: Yes

6. Review Comments to the Author

You may also provide optional suggestions and comments to authors that they might find helpful in planning their study.

Reviewer #2: The title is too long: suggesting to use: facilitators and barriers of relatives involvement in nursing care decisions and self-care of patients with brain injury or tumour: a scoping review protocol.

It is good that the proposed scoping review will be performed following the Joanna Briggs Institute's (JBI) methodology for scoping reviews but still the authors should have a conceptual or theoretical framework to guide them on possible independent variables (barrier and facilitators) and how how the relate to each other.

Despite the authors point argument, i still belief that the researches need to be appraised before using their results in the study. Poorly perform researches should not inform science.

Reviewer #3: Based on the initial manuscript provided and the current one submitted, concerns raised previously have been addressed by the authors. However, there is a need for further editing. For example, Line 125: The objective of this scoping review is to identify and map the available evidence concerning on what kind of…Please review the use of prepositions ‘’concerning on what kind of …Authors should review the use of the preposition ''on'' I think it was a typographical error.

An adequate justification was provided for the scoping review, this is critical. Well done!

Given the use of a structured scoping review framework, the different sections of the proposal were guided by the chosen JBI model, which is good. In terms of feasibility, the proposal is feasible as guided by the JBI model, which was emphasized all through the method section. With this, the scoping review will be structured and organized.

I am particularly impressed with the changes in the method section which makes it easier for readers to understand. The expansion of the search outcome and keywords alongside the mention of the different bibliometric databases authors would use inspires trust and transparency in the work.

Overall, I think the authors have done a good job. They should re-read, review, and edit the typographical errors highlighted above. In my opinion, they have done a great job. Well done!

7. PLOS authors have the option to publish the peer review history of their article (what does this mean?). If published, this will include your full peer review and any attached files.

Reviewer #2: **Yes: **Haddy Tunkara-Bah (Ph. D)

Reviewer #3: **Yes: **Patience James

---

## [Author Response · Author response to Decision Letter 1]

3 Aug 2022

Academic Editor

Robert Didden

Dear Robert Didden,

I hereby send an original research paper entitled Facilitators and barriers of relative’s involvement in nursing care decisions and self-care of patients with acquired brain injury or malignant brain tumour: A scoping review protocol for publication in PLOS ONE. The article is co-authored by Sara Nordentoft, Maria Vilhelmsen, Lena Aadal, Mia Loft and Ingrid Poulsen.

The reviewers made some very useful comments, and we have incorporated their valuable suggestions into the previous version of the manuscript. We hope that this revised manuscript is now suitable for publication in PLOS ONE and look forward to receiving your response.

Sincerely yours

On behalf of the research-group

Rikke Guldager

---

## [Editor Report · Decision Letter 2]

4 Aug 2022

Facilitators and barriers of relatives’ involvement in nursing care decisions and self-care of patients with acquired brain injury or malignant brain tumour: A scoping review protocol.

PONE-D-21-20624R2

Dear Dr. Guldager,

We’re pleased to inform you that your manuscript has been judged scientifically suitable for publication and will be formally accepted for publication once it meets all outstanding technical requirements.

Kind regards,

Robert Didden

Academic Editor

PLOS ONE
---

## [Editor Report · Acceptance letter]

10 Aug 2022

PONE-D-21-20624R2 

Facilitators and barriers of relatives’ involvement in nursing care decisions and self-care of patients with acquired brain injury or malignant brain tumour: A scoping review protocol. 

Dear Dr. Guldager:

I'm pleased to inform you that your manuscript has been deemed suitable for publication in PLOS ONE. Congratulations! Your manuscript is now with our production department. 

Kind regards, 

on behalf of

Professor Robert Didden 

Academic Editor

PLOS ONE